# Searching Strengthens Large Language Models in Finding Bugs of Deep Learning Libraries

## Abstract

Ensuring the quality of deep learning libraries is crucial, as bugs can have significant consequences for downstream software. Fuzzing, a powerful testing method, generates random programs to test software. Generally, effective fuzzing requires generated programs to meet three key criteria: rarity, validity, and variety, among which rarity is most critical for bug detection, as it determines the algorithm's ability to detect bugs. However, current large language model (LLM) based fuzzing approaches struggle to effectively explore the program generation space which results in insufficient rarity and the lack of post-processing leads to a large number of invalid programs and inadequate validity. This paper proposes EvAFuzz, a novel approach that combines Evolutionary Algorithms with LLMs to Fuzz DL libraries. For rarity, EvAFuzz uses a search algorithm to guide LLMs in efficiently exploring the program generation space, iteratively generating increasingly rare programs. For validity, EvAFuzz incorporates a feedback scheme, enabling LLMs to correct invalid programs and achieve high validity. For variety, EvAFuzz constructs a large parent selection space, enriching the diversity of selected parents, and thereby enhancing the variety of generated programs. Our experiments show that EvAFuzz outperforms the previous state-of-the-art (SOTA) in several key metrics. First, in the same version of PyTorch, EvAFuzz detects nine unique crashes, surpassing the SOTA's seven. Next, our method achieves a valid rate of 38.80%, significantly higher than the SOTA's 27.69%. Last, EvAFuzz achieves API coverage rates of 99.49% on PyTorch and 85.76% on TensorFlow, outperforming the SOTA's rates of 86.44% on PyTorch and 69.63% on TensorFlow. These results indicate that our method generates programs with higher rarity, validity, and variety, respectively.

## 1 Introduction

With the advancement of deep learning (DL) technology, DL libraries such as PyTorch(PyTorch) and TensorFlow(TensorFlow) have been widely applied in various fields including scientific research(Jumper et al., 2021; Fawzi et al., 2022), entertainment(Wang et al., 2023a; Silver et al., 2016), and transportation(Yurtsever et al., 2020). However, similarly to other software systems, DL libraries may also harbor security vulnerabilities, which impacts the downstream applications relying on them. To uncover potential errors within DL libraries, an effective approach is to generate a large number of programs to trigger bugs in the libraries. This is known as fuzzing(Odena et al., 2019; Xia et al., 2024; Mansur et al., 2020; Manès et al., 2021). Typically, the effectiveness of fuzzing is influenced by the quality of the generated programs, e.g., rarity, validity, and variety. Compared to regular programs, a rare and valid (namely correct) program is more likely to cover a certain edge case, which leads to a higher probability of triggering bugs. A diverse set of programs can comprehensively cover the code of the library being tested. Therefore, generating programs with these three characteristics is crucial for enhancing the efficiency of fuzzing.

Recently, due to the promising code generation capabilities demonstrated by Large Language Models (LLMs), researchers have begun to explore how to harness these models to generate high-quality programs for fuzzing. Although some methods (Deng et al., 2023; 2024) have already enhanced the efficiency of fuzzing by using code generated by LLMs, relying entirely on LLMs to autonomously generate rare and valid programs remains challenging. There are two main reasons. On one hand, the training data for LLMs primarily consists of common programs that do not easily trigger errors in DL libraries, leading to difficulties for LLMs in generating rare programs that differ from the

training data. On the other hand, because rare programs share similarities with invalid error programs, LLMs need to carefully avoid generating invalid programs while attempting to produce rare ones. Consequently, the LLM-based fuzzing still grapples with the insufficient rarity and validity issue.

To tackle the aforementioned challenges, this paper presents a novel framework that synergistically combines LLMs with searching algorithms. At the heart of this framework lies the Evolutionary Algorithm and large language model based search for Fuzzing (EvAFuzz) algorithm, which employs the search algorithm to guide LLMs to efficiently explore the program generation space, thereby enhancing the rarity of the generated programs. By selecting high-scoring programs as parents and using them as references to produce offspring, EvAFuzz dives deeper into the program generation space, generating increasingly rare programs that cover special edge cases. To mitigate the low validity issue, we propose a feedback scheme, where the execution result of each generated program is fed back to the LLM, enabling it to correct invalid programs and cover more edge cases. Additionally, we construct a large parent selection space, enriching the diversity of selected parents, and thereby enhancing the variety of generated programs. Figure 1 is an overview of our proposed framework.

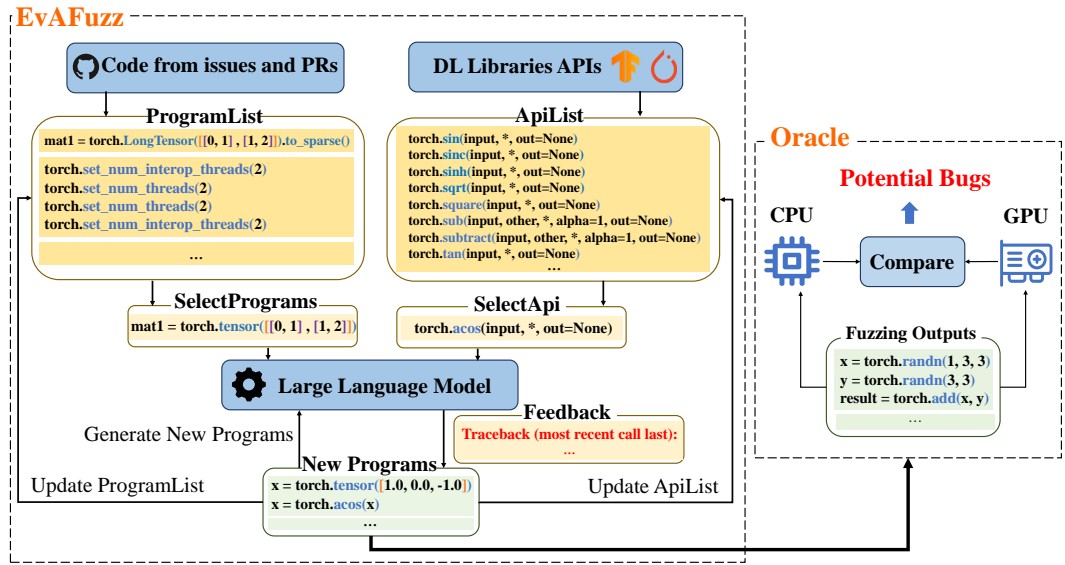

Figure 1: Overview of EvAFuzz.

The contributions of this paper are summarized as follows:

- We propose EvAFuzz, a novel approach that harnesses the power of the Evolutionary Algorithm with LLMs to Fuzz DL libraries. EvAFuzz utilizes a search algorithm to guide LLMs to explore the program generation space for rare programs. The design of EvAFuzz ensures a balance between search depth and breadth for both rarity and variety.
- To enhance validity, we introduce a feedback scheme that feeds the execution results of generated programs back to the LLM, allowing it to correct invalid programs.
- Our comprehensive experiments show the superiority of EvAFuzz, outperforming the state-of-the-art methods in terms of rarity and validity, and comparable in variety. Moreover, our experiments confirm that EvAFuzz successfully searches for increasingly rare programs.

Notably, our approach has discovered several bugs in the current nightly version of PyTorch and TensorFlow. We show some examples in Appendix G.

## 2 BACKGROUND AND RELATED WORK

### 2.1 FUZZING TECHNIQUES

Fuzzing (Liang et al., 2018; Li et al., 2018; Manès et al., 2021) is a software testing technique that involves generating random programs to detect potential security vulnerabilities, bugs, and

crashes. Traditional fuzzers can be categorized into two main types: generation-based (Livinskii et al., 2020; Yang et al., 2011) and mutation-based (Lemieux & Sen, 2018; Zhu et al., 2019). Generation-based fuzzers, also known as grammar-based (Liang et al., 2018) fuzzers, leverage grammar and knowledge of the target language and software semantics to generate complete programs. In contrast, mutation-based fuzzers generate programs by randomly mutating seed programs. Beyond traditional fuzzing approaches, researchers have explored the application of deep learning techniques to develop innovative fuzzing tools.

After generating programs, fuzzers employ an oracle (Wang et al., 2022; 2023b) to execute the generated programs and detect potential bugs in libraries. Oracles are custom-designed for each fuzzing scenario, and currently, there are three primary types of oracles for fuzzing deep learning (DL) libraries: the crash oracle, the consistency oracle (Deng et al., 2023; Wei et al., 2022) and the automatic differentiation (AD) oracle (Yang et al., 2023). The crash oracle detects crashes during program execution. If one occurs, this would be a serious bug. The consistency oracle (Deng et al., 2023; Wei et al., 2022) executes generated programs on diverse backends, such as CPU and GPU, and verifies whether their outputs are consistent. Any inconsistencies detected indicate potential bugs in DL libraries. In contrast, the AD oracle (Yang et al., 2023) leverages first-order and high-order gradients of tensors to determine whether a bug in the DL libraries is triggered.

## 2.2 LLMs FOR FUZZING

LLMs have demonstrated impressive capabilities in generating high-quality code, completing partial code, and even writing entire programs from scratch. This has been achieved by training these models on massive corpora of text data sourced from the Internet, including books, articles, and websites. In contrast to fine-tuning methods, which involve updating the model weights by training on a specific downstream task dataset to create specialized models, in-context learning uses the pre-trained LLM without modifying its weights. Instead, it constructs a prompt that includes multiple examples of input-output demonstrations along with the final task to be solved. TitanFuzz(Deng et al., 2023) first employs Codex(Chen et al., 2021) to generate high-quality seed programs and use InCoder(Fried et al., 2023) to mutate these seed programs. Along these lines, FuzzGPT (Deng et al., 2024) prompt historical buggy programs to LLMs. These works have demonstrated the feasibility of directly utilizing modern LLMs for end-to-end fuzzing of real-world systems without fine-tuning.

## 3 PROPOSED APPROACH

### 3.1 PRELIMINARIES

We first clarify some concepts that will frequently appear in this paper.

*Validity and Invalidity*. Valid programs can run without any errors at least on one backend but may produce inconsistent results on different backends, such as CPU and GPU, due to bugs in libraries. Invalid programs refer to programs that have bugs themselves, such as syntax errors or using undefined variables, and trigger errors during execution. The valid rate refers to the proportion of valid programs relative to all the programs generated.

*Rarity*. A rare program is a program that covers a specific edge case, which typically differs significantly from usual programs and may resemble invalid programs closely. In our algorithm, a program is considered rarer if it is located at a deeper search depth.

*Validity-Rarity Trade-off*. The validity-rarity trade-off refers to the phenomenon where the rarity of generated programs improves at the expense of their validity, making it impossible for both to be high simultaneously. This observation was proposed by (Deng et al., 2024). This principle can aid in analyzing various aspects, such as determining changes in the rarity of generated programs by observing variations in their validity.

### 3.2 EVOLUTIONARY ALGORITHM FRAMEWORK FOR FUZZING (EVAFUZZ) - RARITY

We first describe EvAFuzz in Algorithm 1. The motivation here is based on our observation, i.e., directly prompting a LLM to generate programs is equivalent to a search of depth 1. It means that by simply prompting something like "Please write an unusual program using PyTorch" into a LLM, the

generated results will not meet this requirement. In other words, these generated results will still be very similar to the LLM's training data which mostly consists of correct programs that do not trigger bugs in DL libraries. Therefore, to enhance the ability of LLMs to generate rare programs, we use evolutionary algorithms (EA) for searching and generating increasingly rare programs iteratively. The proposed framework is given in Algorithm 1.

---

**Algorithm 1:** Evolutionary Algorithm For Fuzzing (EvAFuzz)

---

**Input:** Programs of issues and PRs from GitHub, list of tested APIs $ApiList$, target number of generated programs $TargetNum$, the number of seed programs selected at one time $NumPrograms$, retry threshold $MaxRetry$

**Output:** The generated programs

1   Initialize ($ProgramList$, programs of issues and PRs from GitHub)

2   **while** $NumGenerated < TargetNum$ **do**
3      $ApiToGenerate$ = SelectApi($ApiList$)
4      $SeedPrograms$ = SelectPrograms($ProgramList, NumPrograms$)
5      $NewPrograms$ = LLM($ApiToGenerate, SeedPrograms$)
6      **for** $EachProgram\ in\ NewPrograms$ **do**
7         $RepeatCnt = 0$
8         $ExecRes$ = Exec($NewProgram$)
9         **while** $ExecRes\ is\ Failed\ and\ RepeatCnt < MaxRetry$ **do**
10            $FeedbackPrompt$ = ConstructPrompt($ExecRes$)
11            $EachProgram$ = LLM($FeedbackPrompt$)
12            $ExecRes$ = Exec($EachProgram$)
13            $RepeatCnt\ += 1$
14         **end**
15      **end**
16      $Scores$ = FitnessFunc($NewPrograms$)
17      Update ($ApiList, ApiToGenerate, NewPrograms$)
18      Update ($ProgramList, NewPrograms, Scores$)
19      Update ($NumGenerated, NewPrograms$)
20 **end**

21 **return** $ProgramList$

---

The $ApiList$ and $ProgramList$ contain all the APIs provided by the test library and all the generated programs, respectively. The initial $ProgramList$ is constructed with programs from issues and pull requests on GitHub, with each program labeled with the API that triggers the tested library's bugs and the title of the issue or pull request as a bug description. The algorithm begins by selecting an API from the $ApiList$ that is used by the newly generated programs. The goal is to attempt to trigger bugs in the tested library using this API. Next, it selects multiple programs from the $ProgramList$ to serve as the parent programs, i.e., seed programs, for this iteration. The selected API and seed programs are then passed to the Large Language Model (LLM), which generates new offspring programs. The algorithm then enters the feedback stage, where the LLM attempts to correct any invalid programs that were generated. After the feedback stage, the newly generated programs are scored using the $FitnessFunc$. Finally, the $ApiList$, $ProgramList$, and $NumGenerated$ are updated accordingly and the next iteration begins, continuing until the number of generated programs reaches the desired value. **We can see that the search algorithm, i.e., the evolutionary algorithm, guides the LLM in exploring the program generation space, iteratively producing programs with increasing depth and rarity.**

EvAFuzz is based on a few-shot learning approach, leveraging seed programs as exemplars in Algorithm 1 line 5. We input these seeds into the LLM to facilitate learning the intrinsic characteristics of rare programs, enabling the generation of similarly rare, bug-triggering programs. Each seed contains an API declaration, a bug description, and the corresponding program. The LLM learns how the program leverages the API declaration to trigger the described bug, allowing it to generate new programs likely to uncover bugs in the API. Importantly, we use the full API declaration, not just the name, to guide the LLM in learning proper API usage, such as input parameter characteristics. This helps the LLM generate programs that effectively test the target library and trigger vulnerabilities.

The fitness function used in Algorithm 1 line 16 is defined as (Deng et al., 2023), which is used to describe the amount of information contained in a program, i.e., its rarity.

$$FitnessFunc(C) = D + U - R \tag{1}$$

where $C$, $D$, $U$, and $R$ are defined as:

- $C$: A program using tested library's APIs.
- $D$: Depth of dataflow graph[1] which is constructed from $C$. Its edges represent the variable dependencies between two operations in $C$.
- $U$: The number of unique library API calls in $C$.
- $R$: The number of repeated library API calls with the same inputs in $C$.

### 3.3 FEEDBACK SCHEME - VALIDITY

A clear challenge of using LLMs for fuzzing is the validity of generated programs, due to constraints of syntax, semantics, tensor operations, and dimensionality. Previous study FuzzGPT(Deng et al., 2024) shows that the state-of-the-art LLM-generated programs have not exceeded a 30% valid rate. To address this issue, we propose a feedback scheme as shown in Algorithm 1 line 6-15, which feeds the execution results of generated programs back to the LLM, allowing it to correct the programs.

We categorize the issues with invalid programs into two types: exceptions occurring during runtime and the failure to call given APIs. "$ExecRes$ is Failed" in Algorithm 1 line 9 represents that at least one of these two situations occurs. For these two scenarios, we design two corresponding **Feedback Prompts**: **Exception Prompt** and **Not Call Prompt**. The content of **Exception Prompt** and **Not Call Prompt** is explained in detail with an example in Appendix A.

The feedback scheme starts with executing the newly generated program. If it fails, we construct an **Exception Prompt** based on the execution result for the LLM, supplying multi-faceted error information to enable effective program correction. If the program runs successfully but fails to call the specified API, we then construct a **Not Call Prompt** to guide the LLM in modifying the program to call the given API. This iterative process continues until the program runs successfully or the retry limit is met. **Through the feedback scheme, we significantly improve the validity of the generated programs.**

### 3.4 SELECTION STRATEGIES - VARIETY

---
**Algorithm 2:** API Selection

**Input:** List of tested APIs $ApiList$
**Output:** The selected API
1   $NumGeneratedList = [\,]$
2   **for** $API$ in $ApiList$ **do**
3     |   $NumGeneratedList.append(API.NumGenerated)$
4   **end**
5   $p = \texttt{Softmax}(-(NumGeneratedList - \texttt{Avg}(NumGeneratedList)))$
6   $ApiToGenerate = RandomChoice(ApiList, p)$
7   **return** $ApiToGenerate$

---

Below, we will highlight the details of selection strategies for the APIs (Algorithm 2) and the seed programs (Algorithm 3). The core design is to balance rarity and variety, ensuring that the generated programs have high rarity while maximizing variety.

In Algorithm 2, we first retrieve the number of programs generated for each API and construct a list. Due to the large number of generated programs for each API, direct exponentiation would result in precision overflow. To mitigate this, we perform a centralization operation. Since our goal is to assign a higher probability to APIs with fewer generated programs, we take the negative value of

---
[1]We explain the meaning of dataflow graph in the Appendix E

---

**Algorithm 3:** Programs Selection

---

**Input:** List of generated programs $ProgramList$, the number of seed programs selected at one
    time $NumPrograms$
**Output:** The selected seeds

1   $ScoreList = [\,]$
2   **for** $Seed\ in\ ProgramList$ **do**
3     |   $ScoreList.append(Seed.Score)$
4   **end**
5   $p = \text{Softmax}(ScoreList)$
6   $SeedPrograms = RandomChoice(ProgramList, p, NumPrograms)$
7   **return** $SeedPrograms$

---

the centralized result as the input for the $Softmax$ function. The $Softmax$ function subsequently yields the probability distribution for API selection, and we randomly select an API based on this probability distribution. The process of Algorithm 3 is similar, except that the input of Softmax is replaced with the score of each program. From the process of Algorithms 2 and 3, we can observe that during each iteration, the scope of selected APIs and seed programs encompasses the entirety of APIs and previously generated programs.

There are several advantages to these selection strategies. **Firstly, the extensive selection space for seed programs, i.e., parent programs, enhances the diversity of chosen parents, thereby increasing the diversity of generated programs.** Secondly, this approach is in contrast to prior LLM-based fuzzers, which limit seed program selection to those that have the same API as the current selected API. Our approach allows the LLM to learn the intrinsic characteristics of bug-triggering programs, rather than being confined to specific APIs. Lastly, we assign a higher probability of selection to programs with higher scores, which improves the rarity of the generated programs.

## 4   EXPERIMENTS

In the subsequent experiments, we aim to investigate the following problems:

- Can our proposed EvAFuzz outperform the previous state-of-the-art (SOTA) results in terms of the number of detected bugs and coverage on DL libraries?

- Can the evolutionary algorithm successfully guide LLMs to explore the program generation space efficiently, generating programs that are increasingly rare and more likely to trigger bugs in the libraries?

- Whether each component of our proposed EvAFuzz is effective?

- What characteristics do the additional bugs we discover exhibit?

- Does the validity-rarity trade-off hold?

Before delving into the specifics of our experiments, we would like to emphasize that our approach is versatile and not limited to deep learning libraries. The primary reason for choosing deep learning libraries is their significance within the AI ecosystem. In Appendix C, we demonstrate the versatility of our method by conducting experiments on a broader range of libraries.

### 4.1   METRIC

We utilize the following metrics to measure the experimental results:

**Line coverage and API coverage.** The number of lines and APIs, respectively, of internal DL library code that are executed after running the generated programs. The corresponding rates are obtained by dividing by the total number of lines and APIs of the DL library code separately.

**Valid Rate.** It refers to the proportion of valid programs among all the generated programs.

**Crash.** This includes aborts, segmentation fault, and $INTERNAL\_ASSERT\_FAILED$. Crash bugs can potentially lead to critical security issues, and library users are unable to resolve crash bugs through their exception-handling code.

## 4.2 EXPERIMENTS SETUP

**Hyperparameters of LLM inference and EvAFuzz.** We utilize the state-of-the-art code generation model, CodeQWen1.5-7B-Chat. We further explain our rationale for choosing the LLM and conduct experiments on more diverse models in Appendix D. We set $temperature = 0.8$ and $max\_tokens = 1024$. We choose $NumPrograms = 2$, which enables the model to learn the characteristics of rare programs and avoid excessive restriction and leads to generating various programs. We generate five new programs per iteration, and our default setting for $MaxRetry$ is 1.

**Tested libraries.** We focus on fuzzing PyTorch and TensorFlow, the two most widely used deep learning (DL) libraries, consistent with previous testing efforts. For metric calculation, we utilize PyTorch 1.12.1 and TensorFlow 2.10.0, aligning with previous work. To uncover new bugs, we leverage nightly versions of both libraries.

**Environment.** Our experiments are conducted on an Ubuntu 18.04 machine with 8 NVIDIA 3090 GPUs and an Intel(R) Xeon(R) Gold 6246R CPU. We utilize coverage.py(coveragepy) to accurately measure coverage.

**Oracles.** After generating the program, we need oracles to execute the generated programs and determine whether they trigger bugs in the libraries based on the execution results. Similar to (Deng et al., 2023), we employ two types of oracles: the crash oracle and the consistency oracle. The crash oracle detects whether a crash is triggered during program execution, which is the most severe type of bug. The consistency oracle checks whether the program produces inconsistent results across different backends, such as CPU and GPU.

**Baselines.** All the results of the baselines are obtained from their respective papers.

## 4.3 COMPARISON IN TERMS OF RARITY, VALIDITY, AND VARIETY

Firstly, We compare the number of unique detected crash bugs with previous works in Table 1. Following (Deng et al., 2024), we excluded inconsistency bugs from this comparison, as crashes are more straightforward to quantify and can be used as a proxy to evaluate bug detection capabilities. **These results illustrate the rarity of programs generated by the method.** EvAFuzz detects nine unique crashes and outperforms the state-of-the-art (SOTA) FuzzGPT(Deng et al., 2024) which detects seven at most. This indicates the rarity of the generated program of our proposed algorithm and proves that searching strengthens large language models in finding bugs. We list all crash bugs detected by EvAFuzz in Appendix F.

Table 1: (Rarity) Comparing the number of unique crashes with previous works.

| | TitanFuzz(Deng et al., 2023) | FuzzGPT(Deng et al., 2024) | | | EvAFuzz(Ours) |
|---|---|---|---|---|---|
| | | Few Shot | Zero Shot | Fine Tune | |
| Crashes | 3 | 7 | 7 | 2 | **9** |

Secondly, We compare the valid rate of the generated programs with previous LLM-based approaches(Deng et al., 2023; 2024) in Table 2. **These results demonstrate the validity of programs generated by the method.** Notably, EvAFuzz achieves a valid rate of up to 38.8% on PyTorch and 34.04% on TensorFlow, respectively, outperforming the SOTA TitanFuzz(Deng et al., 2023) results of 38.2% on PyTorch and 30.67% on TensorFlow. According to the validity-rarity trade-off, the low number of crash bugs detected by TitanFuzz(Deng et al., 2023) implies that the generated programs lack sufficient rarity, leading to their high validity. However, even so, the validity of the programs generated by TitanFuzz(Deng et al., 2023) is not as good as that of our EvAFuzz. This improvement underscores the significant effectiveness of our feedback scheme in generating valid programs.

Finally, we compare the line coverage and API coverage with several SOTA DL library fuzzers in Table 3. **These results indicate the variety of programs generated by the method.** Our proposed

Table 2: (Validity) Comparison of valid rate with previous LLM-based fuzzers. The numbers in the Valid and All columns in the table represent the number of generated programs.

| | Method | Valid | All | Valid Rate(%) |
|---|---|---|---|---|
| PyTorch | TitanFuzz(Deng et al., 2023) | 6969 | 18245 | 38.20% |
| | FuzzGPT-FS(Deng et al., 2024) | 42496 | 154904 | 27.43% |
| | FuzzGPT-ZS(Deng et al., 2024) | 7809 | 132111 | 5.91% |
| | FuzzGPT-FT(Deng et al., 2024) | 31225 | 112765 | 27.69% |
| | EvAFuzz(Ours) | 47574 | 122612 | **38.80%** |
| TensorFlow | TitanFuzz(Deng et al., 2023) | 5173 | 16865 | 30.67% |
| | FuzzGPT-FS(Deng et al., 2024) | 54058 | 310483 | 17.41% |
| | FuzzGPT-ZS(Deng et al., 2024) | 4650 | 233887 | 1.99% |
| | FuzzGPT-FT(Deng et al., 2024) | 31105 | 253216 | 12.28% |
| | EvAFuzz(Ours) | 20187 | 59308 | **34.04%** |

EvAFuzz achieves a line coverage rate of 29.66% on PyTorch and 47.48% on TensorFlow, along with an API coverage rate of 99.49% on PyTorch and 85.76% on TensorFlow. These results outperform the SOTA FuzzGPT-Few Shot(Deng et al., 2024), which attains API coverage rates of 86.44% on PyTorch and 69.63% on TensorFlow. This indicates that the variety of programs generated by our method is comparable to that of the SOTA method.

Table 3: (Variety) Comparison on coverage with previous works.

| | PyTorch | | TensorFlow | |
|---|---|---|---|---|
| | Line Coverage | API Coverage | Line Coverage | API Coverage |
| Codebase Under Test | 113538(100%) | 1593(100%) | 269448(100%) | 3316(100%) |
| FreeFuzz(Wei et al., 2022) | 15688(13.82%) | 468(29.38%) | 78548(29.15%) | 581(17.52%) |
| DeepREL(Deng et al., 2022) | 15794(13.91%) | 1071(67.23%) | 82592(30.65%) | 1159(34.95%) |
| $\nabla$Fuzz(Yang et al., 2023) | 15860(13.97%) | 1071(67.23%) | 89722(33.3%) | 1159(34.95%) |
| Muffin(Gu et al., 2022) | NA | NA | 79283(29.42%) | 79(2.38%) |
| TitanFuzz(Deng et al., 2023) | 23823(20.98%) | 1329(83.43%) | 107685(39.97%) | 2215(66.80%) |
| FuzzGPT-Few Shot(Deng et al., 2024) | 35426(31.2%) | 1377(86.44%) | **146487(54.37%)** | 2309(69.63%) |
| FuzzGPT-Zero Shot(Deng et al., 2024) | **38284(33.72%)** | 1237(77.65%) | 126193(46.83%) | 1460(44.03%) |
| FuzzGPT-Fine Tune(Deng et al., 2024) | 36463(32.12%) | 1223(77.65%) | 125832(46.70%) | 1834(55.31%) |
| EvAFuzz(Ours) | 33678(29.66%) | **1585(99.49%)** | 127953(47.48%) | **2844(85.77%)** |

## 4.4 ALGORITHMIC ANALYSIS

We want to explore whether there is a discernible trend in the relationship between the generated programs and their corresponding scores as the search progresses. In other words, we aimed to determine if the rarity of the generated programs, as measured by their scores, continues to improve over the search progress. To investigate this, we plot the average scores of the generated programs at intervals of 2000 against their IDs(the later the program is generated, the larger its ID), as shown in Figure 2(red). The results indicate that, in general, the programs generated later in the search process tend to have higher scores. This suggests that as the search progresses, the generated programs have increasing rarity. This observation aligns with our expectation that the search mechanism is effectively exploring the program generation space and generating programs with higher scores over time.

Additionally, we analyze the valid rates of the generated programs at intervals of 2000 and plot the trend in Figure 2(blue). The graph reveals a decline in valid rates as the search progresses. According to the validity-rarity trade-off, this phenomenon also indicates that the generated programs become increasingly rare, thereby validating the efficacy of our search algorithm.

We further analyze the line coverage trend against the generated program IDs in Appendix B.

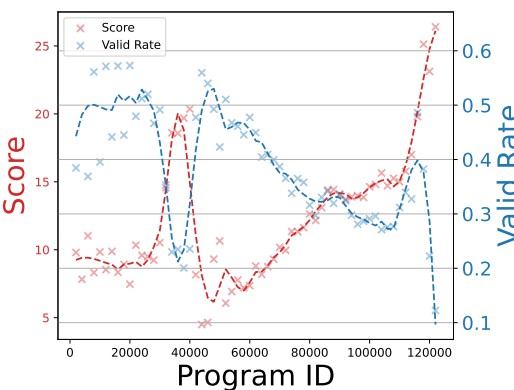 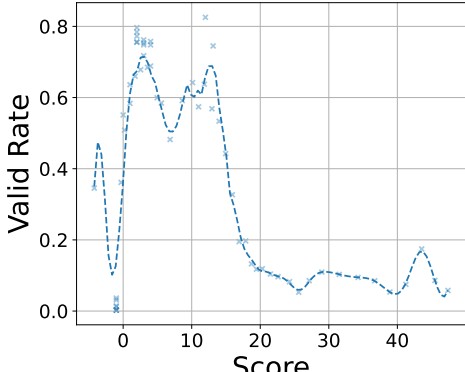

Figure 2: Relationship between score(red)/valid rate(blue) and generated program IDs.

Figure 3: Relationship between valid rate and score (validity-rarity trade-off).

## 4.5 ABLATION STUDY

In this section, we will evaluate the effectiveness of each component of our proposed EvAFuzz algorithm.

**Feedback Scheme**. We set the $NumGenerated$ parameter of Algorithm 1 to 5,000, generating programs to test PyTorch both with and without the feedback scheme. The results are presented in Table 4, which compares the valid rate, API coverage, and line coverage achieved with and without the feedback scheme. As the table demonstrates, all three evaluation metrics - valid rate, API coverage, and line coverage - are significantly improved when feedback is incorporated. This underscores the effectiveness of the feedback mechanism in enhancing the overall validity and variety of the generated programs. Notably, the "Corrected" column indicates the rate of initially invalid programs that were successfully corrected to be valid through the feedback process. We can observe that the $valid\ rate - corrected\ rate$ of w/ feedback is greater than the $valid\ rate$ of w/o feedback. We analyze that the feedback scheme increases the proportion of valid programs selected as few-shot examples, thereby reinforcing the generation of more valid programs. However, without the feedback scheme, invalid programs dominate as seed programs, increasing the likelihood of generating more invalid programs.

Table 4: EvaFuzz w/ or w/o feedback scheme.

|  | Valid Rate(%) | Corrected Rate(%) | API Coverage | Line Coverage |
|---|---|---|---|---|
| w/ feedback | 62.66% | 16.01% | 902(56.62%) | 27660(24.36%) |
| w/o feedback | 17.02% | NA | 312(19.59%) | 24372(21.47%) |

**Selection Strategies**. We further evaluate the strategies for selecting APIs and seeds in Table 5, using uniform random selection as the baseline. The three columns in the table refer to the valid rate, API coverage, and line coverage, respectively. First, let's compare the results between Full and UniformRandomSeeds. UniformRandomSeeds has a higher valid rate, which, according to the validity-rarity trade-off, suggests that the generated programs lack rarity. Meanwhile, its high API coverage indicates better variety. However, we prioritize having high rarity over validity and variety for programs as our primary goal is generating bug-triggering programs. Next, we compare the results between Full and UniformRandomAPI. The much lower API coverage of UniformRandomAPI indicates that the distribution of selected APIs is not uniform under this API selection strategy. We hope to comprehensively test each API, so variety is the top priority when selecting APIs. These results demonstrate that our designed API and seed program selection strategies effectively balance the rarity and variety, achieving maximum rarity while maintaining variety.

In summary, these results fully demonstrate the effectiveness of our designed feedback scheme and selection strategies, enhancing the validity of our EvAFuzz generated programs while balancing both rarity and variety.

Table 5: EvaFuzz with different selection strategy of APIs and seed programs.

| | Valid Rate(%) | API Coverage | Line Coverage |
|---|---|---|---|
| Full | 62.66% | 902(56.62%) | 27660(24.36%) |
| UniformRandomSeeds | 67.56% | 1010(63.40%) | 25939(22.85%) |
| UniformRandomAPI | 50.81% | 646(40.55%) | 25316(22.30%) |

## 4.6 DETECTED BUGS

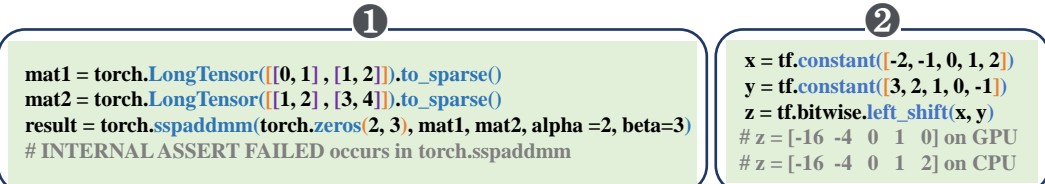

Figure 4: Example bugs found by EvAFuzz.

Figure 4 presents two examples of bugs that we discovered in the current nightly version of PyTorch and TensorFlow - the one on the left is from PyTorch, while the one on the right is from TensorFlow. The `INTERNAL_ASSERT_FAILED` (a crash bug) occurs in the `torch.sspaddmm` module, which is a fundamental component in the computation of sparse tensors used in both transformers and LLMs. The TensorFlow bug, on the other hand, is found in the `tf.bitwise.left_shift` operation, another basic function employed in novel designs such as mask and sparse attentions. `z = [-16, -4, 0, 1, 2]` on CPU but `z = [-16, -4, 0, 1, 0]` on GPU, which is inconsistent. These two examples further demonstrate the effectiveness of our system in uncovering additional bugs beyond what previous approaches had identified. We show more detected bugs in Appendix G.

## 4.7 VALIDITY-RARITY TRADE-OFF

Finally, we want to verify whether the validity-rarity trade-off holds through experiments. This phenomenon can be theoretically attributed to two key factors: rare programs diverge significantly from the training data of LLMs leading to an out-of-distribution problem, and they often bear similarities to invalid programs, making them more likely to generate invalid programs when attempting to generate rare ones. To empirically validate this, we calculate the valid rates and average scores of the generated programs at intervals of 2000, and then draw them in order of increasing scores, as depicted in Figure 3. The results confirm two crucial findings: firstly, the validity-rarity trade-off is a real and existing phenomenon, and secondly, our $FitnessFunc$ effectively captures the rarity of programs. Notably, our experiment reveals an intriguing anomaly: the valid rate exhibits a rapid increase as the score increases when the score is below 3, contradicting the expected validity-rarity trade-off. This suggests that certain short and seemingly common programs can also trigger bugs in the library, implying that the score based on $FitnessFunc$ and rarity are not perfectly correlated, but rather exhibit a certain degree of divergence.

## 5 CONCLUSION

We propose EvAFuzz, a novel fuzzing approach that combines evolutionary algorithms and large language models to search for rare programs in deep learning libraries. Our experiments demonstrate that our proposed EvAFuzz outperforms state-of-the-art methods in terms of rarity and validity, and achieves comparable variety. The extra bugs detected by EvAFuzz root in basic computations on sparse matrices and bitwise left shift operations resulted in precision bugs in modern transformers and LLMs. This highlights the effectiveness of the EvAFuzz approach in leveraging the power of search algorithms to strengthen LLMs in finding bugs of deep learning libraries.

## 6 REPRODUCIBILITY

We provide all experiment setups of our method in Section 4.2. The programs used for initializing the $ProgramList$ in our paper come from the issues and PRs of PyTorch(PyTorch) and Tensor-Flow(TensorFlow). We use the oracles from (Deng et al., 2023). The LLM CodeQWen1.5-7B-Chat we use is an open-source model, which can be obtained from HuggingFace(HuggingFace). The evolutionary algorithm we use is a well-established algorithm and is easy to reproduce. We need to organize the experiment code and write documentation, which will be made publicly available as soon as possible.

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

## A  FEEDBACK PROMPT

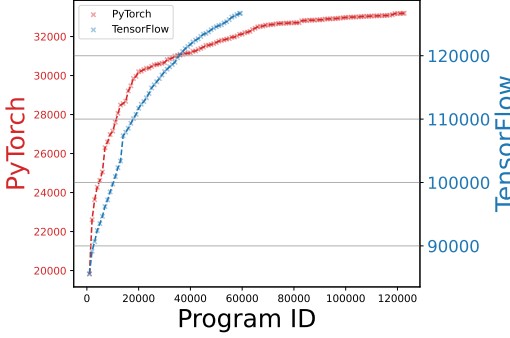

Figure 5: Two examples of **Feedback Prompt**, namely **Exception Prompt**(left) and **Not Call Prompt**(right).

In this section, we will explain the details of the **Feedback Prompt** including **Exception Prompt** and **Not Call Prompt**. Figure 5 shows two instances.

The **Exception Prompt** consists of four parts. *Task Description*: Inform the LLM that it is receiving a program with errors and instruct it to correct the program. *Error Code*: The faulty program that needs correction. *Exception Type*: The types of errors that occur during execution, including $SyntaxError$, $RuntimeError$, etc. *Exception Info*: The detailed error information during execution, specifically the runtime traceback that appears after an error occurs.

The **Not Call Prompt** includes two parts. *Task Description*: Inform the LLM that it will receive a program and instruct it to rewrite the program to call the given API, with the description of the given API. *Code*: The program that needs to be modified.

## B  LINE COVERAGE TREND

Figure 6: Line coverage trend of PyTorch(red) and TensorFlow(blue).

We show the line coverage trend of the EvAFuzz generated programs in this section. Figure 6 is the change in line coverage with the increase in the number of generated programs. We can observe that the trend remains consistently upward, and maintaining this trend even at the peak program counts(PyTorch: 122,611, TensorFlow: 59,307). This suggests that generating additional programs would likely further improve line coverage.

## C  FINDING BUGS IN MORE THAN DEEP LEARNING LIBRARIES

Table 6: Apply our proposed EvAFuzz on more than deep learning libraries including NumPy(1.22.3) and SciPy(1.10.1).

|  | NumPy | | | SciPy | | |
|---|---|---|---|---|---|---|
|  | Line Coverage | API Coverage | Crash Bugs | Line Coverage | API Coverage | Crash Bugs |
| Code Under Test | 106381(100%) | 1293(100%) | NA | 67139(100%) | 1733(100%) | NA |
| EvAFuzz(Ours) | 18149(17.06%) | 1289(99.69%) | 1 | 28451(42.38%) | 1726(99.60%) | 1 |

In this section, we conducte experiments on more libraries to demonstrate the generalization of our approach, which is not limited to deep learning libraries. Specifically, we experiment on NumPy 1.22.3 and SciPy 1.10.1, generating 21,430 and 12,539 programs, respectively. The results are presented in Table 6. For instance, our method detects **1 crash bug** on SciPy, achieving **99.60% API coverage** and **42.38% line coverage**. These results demonstrate that our proposed EvAFuzz is generalizable and applicable to non-DL libraries.

# D EXPERIMENTS WITH DIFFERENT LLMS

Table 7: Line coverage and API coverage using other LLMs, including deepseek coder-7b-instruct-v1.5, Llama 3-8B-Instruct, and Nxcode-CQ-7B-orpo.

|  | PyTorch | | TensorFlow | |
| --- | --- | --- | --- | --- |
|  | Line Coverage | API Coverage | Line Coverage | API Coverage |
| Codebase Under Test | 113538(100%) | 1593(100%) | 269448(100%) | 3316(100%) |
| deepseek coder-7b-instruct-v1.5 | 29422(25.19%) | 1568(98.43%) | 98239(36.46%) | 3113(93.88%) |
| Llama 3-8B-Instruct | 27435(24.16%) | 1541(96.74%) | 92685(34.40%) | 947(28.56%) |
| Nxcode-CQ-7B-orpo | 28249(24.88%) | 1562(98.05%) | 101570(37.70%) | 2918(88.00%) |
| CodeQwen1.5-7B-Chat | 26949(23.74%) | 1585(99,05%) | 99756(37.02%) | 2844(85.77%) |

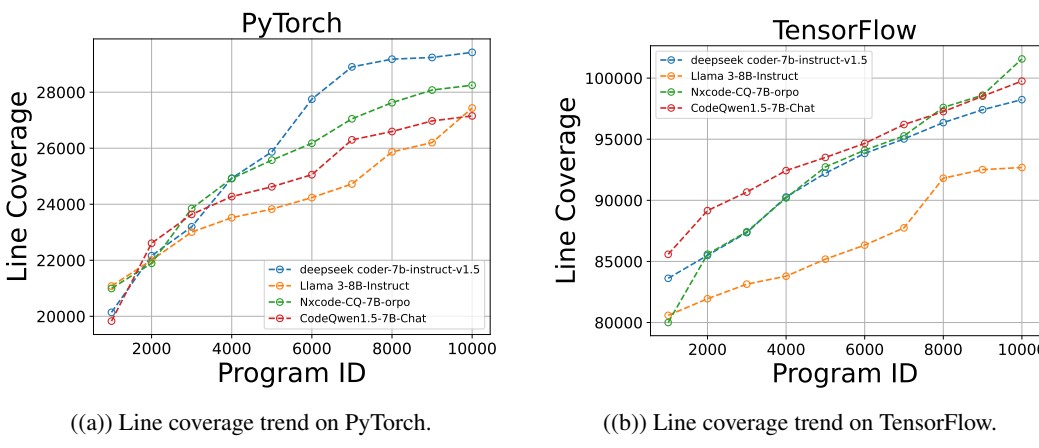

((a)) Line coverage trend on PyTorch.  ((b)) Line coverage trend on TensorFlow.

Figure 7: Trend of line coverage and API coverage using other LLMs.

In this section, we explain the basis for our selection of the LLM and conduct experiments on more LLMs.

We selected CodeQWen1.5-7B-Chat for our experiments because it was the state-of-the-art code generation Large Language Model (LLM) at the time of our experiments, as indicated by the BigCode leaderboard on HuggingFace.

Additionally, we conducte more experiments on models of similar size. We generate 10,000 programs each for PyTorch and TensorFlow using DeepSeek Coder-7b-instruct-v1.5, Llama 3-8B-Instruct, and Nxcode-CQ-7B-orpo, and compared them with CodeQwen1.5-7B-Chat. Table 7 shows the results, and Figure 7 illustrates the line coverage trend for different models on PyTorch and TensorFlow as the number of generated programs increases. We can notice that except for Llama3, which performs poorly on TensorFlow, the performance of the different models was generally similar. We want to emphasize that our approach is not tied to a specific model. Naturally, the stronger the model's performance, the better the results.

# E    DATAFLOW GRAPH

The dataflow graph (DFG) is a concept of compilation. It is a graph that represents data dependencies between a number of operations, e.g., a dataflow graph $a \to + \leftarrow b$ is related to $a + b$. We show an example in Figure 8. Each node of the graph is an input or a calculation result. Each edge of the graph is the calculation dependency.

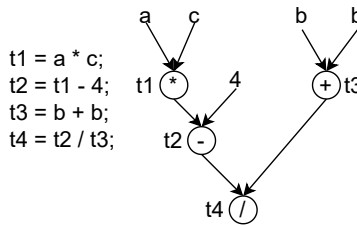

```
t1 = a * c;
t2 = t1 - 4;
t3 = b + b;
t4 = t2 / t3;
```

Figure 8: An example of dataflow graph.

# F    CRASH BUGS

In this section, we list all crash bugs detected by EvAFuzz in PyTorch version 1.12.1+cu113.

Listing 1: Crash Bug 1

```
"""
error : segmentation  fault
"""
import torch
import torch.distributed as dist
# Initialize  the  distributed  environment
dist.init_process_group("gloo", init_method="tcp://localhost:12345", rank
    =0, world_size=1)
# Create  a tensor
tensor = torch.tensor([1, 2, 3, 4])
# Send the  tensor  to rank 0
dist.send(tensor, dst=0)
# Finalize  the  distributed  environment
dist.destroy_process_group()
```

Listing 2: Crash Bug 2

```
"""
terminate  called  after  throwing  an  instance  of  'c10::Error'
  what():  Error: cannot  set  number of interop  threads  after  parallel  work has  started
      or  set_num_interop_threads  called
Exception  raised  from  set_num_interop_threads  at  ../ aten/ src /ATen/
    ParallelThreadPoolNative .cpp:54 (most  recent  call  first ):
frame #0: c10::Error::Error(c10::SourceLocation,  std :: string )  + 0x3e (0x7fe43af9520e in
    .../ lib /python3.8/ site −packages/torch/ lib / libc10 .so)
frame #1: c10:: detail :: torchCheckFail(char const ∗, char const ∗, unsigned  int ,  char const
    ∗) + 0x60 (0x7fe43af706a9 in  .../ lib /python3.8/ site −packages/torch/ lib / libc10 .so)
frame #2: <unknown function> + 0x178d38f (0x7fe464ea338f in  .../ lib /python3.8/ site −
    packages/torch/ lib / libtorch_cpu .so)
frame #3: <unknown function> + 0x5e9dfa (0x7fe48cd44dfa in  .../ lib /python3.8/ site −
    packages/torch/ lib / libtorch_python .so)
frame #4: python() [0x4e76fb]
<omitting  python frames>
frame #9: python() [0x5a6541]
frame #10: python() [0x5a554f]
```

```
frame #11: python() [0x45c485]
frame #13: python() [0x44fb81]
frame #15: __libc_start_main + 0xe7 (0x7fe4a8294bf7 in /lib/x86_64-linux-gnu/libc.so.6)
frame #16: python() [0x57a64d]
"""
import torch
torch.get_num_interop_threads()
torch.get_num_threads()
torch.set_num_threads(1)
torch.set_num_threads(4)
torch.set_num_interop_threads(1)
torch.set_num_interop_threads(4)
```

Listing 3: Crash Bug 3

```
"""
double free or corruption (out)
"""
import torch
LU_data = torch.tensor([[1, 2], [3, 4]], dtype=torch.float32)
LU_pivots = torch.tensor([0, 1], dtype=torch.int32)
b = torch.tensor([[5], [6]], dtype=torch.float32)
torch.lu_solve(b, LU_data, LU_pivots)
```

Listing 4: Crash Bug 4

```
"""
error: segmentation fault
"""
import torch
input = torch.randn(2, 1, 5, 5, 5)
kernel_size = (1, 2, 2)
stride = (1, 2, 2)
ceil_mode = False
output_size = (1, 3, 3, 3, 3)
indices = torch.empty(0)
output = torch.nn.functional.fractional_max_pool3d(input,
    kernel_size, stride, ceil_mode, output_size, indices)
print(output)
```

Listing 5: Crash Bug 5

```
"""
error: segmentation fault
"""
import torch
from torch.overrides import has_torch_function,
    has_torch_function_unary, has_torch_function_variadic
a = torch.tensor(1)
has_torch_function(a)
has_torch_function_unary(a)
has_torch_function_variadic(a)
```

Listing 6: Crash Bug 6

```
"""
error: segmentation fault
"""
import torch
import torch.overrides as overrides
def foo(self, x):
```

```
918        pass
919    # Check if 'foo' overrides a Tensor property or method
920    is_overridden = overrides.is_tensor_method_or_property(foo)
921    print(is_overridden) # False
922    class MyTensor(torch.Tensor):
923        pass
924    @overrides.has_torch_function(MyTensor)
925    def foo(self, x):
926        pass
927    # Check if 'foo' overrides a Tensor property or method
928    is_overridden = overrides.is_tensor_method_or_property(foo)
929    print(is_overridden) # True
```

Listing 7: Crash Bug 7

```
931    """
932
933    error: segmentation fault
934    """
935    import torch
936    def main():
937        torch.manual_seed(0)
938        torch.set_num_threads(1)
939        torch.set_num_interop_threads(1)
940        device = torch.device("cuda" if torch.cuda.is_available() else "
941            cpu")
942        @torch.overrides.wrap_torch_function(torch.sum)
943        def custom_sum(func, input):
944            return torch.sum(input) * 2
945        input = torch.randn(2, 3, device=device) # Move input tensor to the same
946            device as the model parameters
947        output = custom_sum(input)
948        print("All tests passed.")
949    if __name__ == "__main__":
950        main()
```

Listing 8: Crash Bug 8

```
952    """
953    Segmentation fault (core dumped)
954    """
955    import torch
956    def main():
957        torch.manual_seed(0)
958        torch.set_num_threads(1)
959        torch.set_num_interop_threads(1)
960        device = torch.device("cuda" if torch.cuda.is_available() else "
961            cpu")
962        @torch.overrides.wrap_torch_function(torch.sum)
963        def custom_sum(func, input):
964            return torch.sum(input) * 2
965        input = torch.randn(2, 3, device=device) # Move input tensor to the same
966            device as the model parameters
967        output = custom_sum(input)
968        print("All tests passed.")
969    if __name__ == "__main__":
970        main()
```

Listing 9: Crash Bug 9

```
971    """
```

```
972    terminate called after throwing an instance of 'c10::Error'
973      what():  Error: cannot set number of interop threads after parallel work has started
974        or set_num_interop_threads called
975    Exception raised from set_num_interop_threads at ../aten/src/ATen/
976        ParallelThreadPoolNative.cpp:54 (most recent call first):
977    frame #0: c10::Error::Error(c10::SourceLocation, std::string) + 0x3e (0x7ff1bd3ad20e in
978        .../lib/python3.8/site-packages/torch/lib/libc10.so)
979    frame #1: c10::detail::torchCheckFail(char const*, char const*, unsigned int, char const
980        *) + 0x60 (0x7ff1bd3886a9 in .../lib/python3.8/site-packages/torch/lib/libc10.so)
981    frame #2: <unknown function> + 0x178d38f (0x7ff1e72bb38f in .../lib/python3.8/site-
982        packages/torch/lib/libtorch_cpu.so)
983    frame #3: <unknown function> + 0x5e9dfa (0x7ff20f15cdfa in .../lib/python3.8/site-
        packages/torch/lib/libtorch_python.so)
984    frame #4: python() [0x4e76fb]
985    <omitting python frames>
986    frame #6: python() [0x4e4bd2]
987    frame #7: python() [0x5978d2]
988    frame #8: python() [0x5b9cc2]
989    frame #10: python() [0x4e8b8b]
990    frame #18: python() [0x5a6541]
991    frame #19: python() [0x5a554f]
992    frame #20: python() [0x45c485]
993    frame #22: python() [0x44fb81]
994    frame #24: __libc_start_main + 0xe7 (0x7ff22a68bbf7 in /lib/x86_64-linux-gnu/libc.so.6)
995    frame #25: python() [0x57a64d]
       """
996    import torch
997    from torch.utils.data import Sampler
998    class RandomSampler(Sampler):
999        def __init__(self, data_source):
1000            self.data_source = data_source
1001            self.indices = list(range(len(data_source)))
1002        def __iter__(self):
1003            torch.manual_seed(0)
1004            torch.set_num_threads(1)
1005            torch.set_num_interop_threads(1)
1006            torch.shuffle(self.indices)
1007            return iter(self.indices)
1008        def __len__(self):
1009            return len(self.data_source)
1010    def main():
1011        torch.manual_seed(0)
1012        torch.set_num_threads(1)
1013        torch.set_num_interop_threads(1)
1014        device = torch.device("cuda" if torch.cuda.is_available() else "
               cpu")
1015        class MyDataset(torch.utils.data.Dataset):
1016            def __len__(self):
1017                return 10
1018            def __getitem__(self, index):
1019                return torch.randn(1, device=device), torch.randn(1,
                       device=device)
1020        dataset = MyDataset()
1021        sampler = RandomSampler(dataset)
1022        for _ in range(10):
1023            sample = next(iter(sampler))
1024            print(sample)
1025        print("All tests passed.")
       if __name__ == "__main__":
```

```
1026      main()
1027
1028
```

## G  DETECTED BUGS IN NIGHTLY VERSION OF PYTORCH AND TENSORFLOW

In these section, we show several detected bugs in nightly version of PyTorch and TensorFlow.

Listing 10: New Detected Bug 1

```
"""
API: torch.sspaddmm
Exception Type: CpuCrashCatch(420)
Error Message: RuntimeError self.is_sparse() INTERNAL ASSERT FAILED at "../aten/src/
    ATen/native/SparseTensorUtils.h":28, please report a bug to PyTorch.
    _internal_get_SparseTensorImpl: not a sparse tensor
"""
import torch
a = torch.randn(3, 3)
b = torch.randn(3, 3)
c = torch.randn(3, 3)
# Convert tensors to sparse tensors
a_sparse = a.to_sparse()
b_sparse = b.to_sparse()
# Perform sparse matrix multiplication
torch.sspaddmm(c, a_sparse, b_sparse, beta=2.5, alpha=0.1)
torch.sspaddmm(c, a_sparse, b_sparse, beta=1.5, alpha=0.3)
torch.sspaddmm(c, a_sparse, b_sparse, beta=0.75, alpha=0.6)
```

Listing 11: New Detected Bug 2

```
"""
API: torch.QUInt4x2Storage
Exception Type: GpuCrashCatch(420)
Error Message: RuntimeError cuda_dispatch_ptr INTERNAL ASSERT FAILED at "../aten/src/
    ATen/native/DispatchStub.cpp":137, please report a bug to PyTorch. DispatchStub:
    missing CUDA kernel
"""
import torch
def quantize(tensor, scale, zero_point):
    return torch.quantize_per_tensor(tensor, scale, zero_point,
        torch.quint4x2)
tensor = torch.tensor([[[[-0.1, -0.2], [-0.3, -0.4]], [[-0.5,
    -0.6], [-0.7, -0.8]]], [[[0.1, 0.2], [0.3, 0.4]], [[0.5, 0.6],
    [0.7, 0.8]]]])
tensor = tensor.float()  # Convert to float tensor
scale = 0.5 # Define the scale
zero_point = 0 # Define the zero point
quantized_tensor = quantize(tensor, scale, zero_point)
print(quantized_tensor)
```

Listing 12: New Detected Bug 3

```
"""
API: torch.onnx.is_in
Exception Type: CpuCrashCatch(420)
Error Message: RuntimeError: 0 INTERNAL ASSERT FAILED at "../torch/csrc/jit/ir/
    alias_analysis.cpp":608, please report a bug to PyTorch. We don't have an op for
    aten::mul but it isn't a special case. Argument types: Tensor, bool,
Candidates:
    aten::mul.Tensor(Tensor self, Tensor other) -> (Tensor)
    aten::mul.Scalar(Tensor self, Scalar other) -> (Tensor)
```

```
aten :: mul.out(Tensor self , Tensor other , *, Tensor(a!) out) −> (Tensor(a!))
aten :: mul.Scalar_out(Tensor self , Scalar other , *, Tensor(a!) out) −> (Tensor(a
    !))
aten :: mul. left_t (t[] l , int n) −> (t[])
aten :: mul. right_ (int n, t[] l) −> (t[])
aten :: mul. int ( int a, int b) −> (int )
aten :: mul.complex(complex a, complex b) −> (complex)
aten :: mul. float ( float a, float b) −> ( float )
aten :: mul.int_complex( int a, complex b) −> (complex)
aten :: mul.complex_int(complex a, int b) −> (complex)
aten :: mul.float_complex ( float a, complex b) −> (complex)
aten :: mul. complex_float (complex a, float b) −> (complex)
aten :: mul. int_float ( int a, float b) −> ( float )
aten :: mul. float_int ( float a, int b) −> ( float )
aten :: mul(Scalar a, Scalar b) −> (Scalar)
"""
import torch
import torch.onnx
class MyModel(torch.nn.Module):
    def forward(self, x):
        return x * torch.onnx.is_in_onnx_export()
torch.manual_seed(0)
model = MyModel().cuda().eval()
x = torch.tensor([[0.1, 0.2]], device='cuda', dtype=torch.float32)
torch.onnx.export(model, (x, ), "test_is_in_onnx_export.onnx")
```

Listing 13: New Detected Bug 4

```
"""
API: torch .QUInt4x2Storage
Exception Type: GpuCrashCatch(420)
Error Message: RuntimeError cuda_dispatch_ptr INTERNAL ASSERT FAILED at "../aten/src/
    ATen/native/DispatchStub.cpp":137, please report a bug to PyTorch. DispatchStub:
    missing CUDA kernel
"""
import torch
def quantize(tensor, scale, zero_point):
    return torch.quantize_per_tensor(tensor, scale, zero_point,
        torch.quint4x2)
tensor = torch.tensor([[[[-0.1, -0.2], [-0.3, -0.4]], [[-0.5,
    -0.6], [-0.7, -0.8]]], [[[0.1, 0.2], [0.3, 0.4]], [[0.5, 0.6],
    [0.7, 0.8]]]])
tensor = tensor.float() # Convert to float tensor
scale = 0.5 # Define the scale
zero_point = 0 # Define the zero point
quantized_tensor = quantize(tensor, scale, zero_point)
print(quantized_tensor)
```

Listing 14: New Detected Bug 5

```
"""
API: tf . bitwise . left
Origin Path: SummaryResults_20240522_133114_Full/tf/valid/ tf . bitwise . left_shift_2844 .py
Exception Type: VarInconsistentCatch (420)
Error Message:
z on CPU: [−16 −4 0 1 2]
z on GPU: [−16 −4 0 1 0]
"""
import tensorflow as tf
x = tf.constant([-2, -1, 0, 1, 2])
```

```
1134    y = tf.constant([3, 2, 1, 0, -1])
1135    z = tf.bitwise.left_shift(x, y)
1136
1137
```

Listing 15: New Detected Bug 6

```
1138    """
1139    API: torch.Tensor.msort
1140    Exception Type: VarInconsistentCatch (420)
1141    Error Message:
1142     diff :[' indices ']
1143    CPU: 'indices': tensor([ 1,  3,  6,  0,  9,  2,  4,  8, 10,  7,  5]),
1144    GPU: 'indices': tensor([ 1,  3,  6,  9,  0,  2, 10,  8,  4,  7,  5], device='cuda:0')
1145    """
1146    import torch
1147    # Create a tensor
1148    x = torch.tensor([3, 1, 4, 1, 5, 9, 2, 6, 5, 3, 5])
1149    # Sort the tensor
1150    sorted_tensor, indices = torch.sort(x)
1151    print(sorted_tensor)
1152    print(indices)
1153
```

Listing 16: New Detected Bug 7

```
1155    """
1156    API: torch.cross
1157    Exception Type: VarInconsistentCatch (420)
1158    Error Message:
1159    Result on CPU:
1160    tensor([[-0.8090, -1.8866, -0.1531,  0.9241],
1161            [ 0.9158,  2.3039,  0.7830, -0.6937],
1162            [ 1.2427, -3.2395,  0.9753, -1.3094]])
1163    Result on GPU:
1164    tensor([[-0.8090, -1.8866, -0.1531,  0.9241],
1165            [ 0.0454, -0.0165,  2.6868, -0.4181],
1166            [-1.8018,  1.4298,  2.3755, -0.9607]], device='cuda:0')
1167    """
1168    import torch
1169    torch.manual_seed(0)
1170    torch.cuda.manual_seed(0)
1171    a = torch.randn(3, 4)
1172    b = torch.randn(3, 4)
1173    torch.cross(a, b, out=a)
```

Listing 17: New Detected Bug 8

```
1175    """
1176    Testcase ID: 13286
1177    API: torch.empty
1178    Origin Path: SummaryResults_20240522_133114_Full/torch/valid/torch.empty_strided_22538.
1179        py
1180    Exception Type: VarInconsistentCatch (420)
1181    Error Message:
1182     diff :[   'PassFlattenCallTempVar1',
1183        'PassLogTorchIntermediateTempVar1_PassFlattenCallTempVar1']
1184    CPU:
1185    {   'PassFlattenCallTempVar1': tensor([[ 1.1649e-15,  7.8107e-02,  1.5172e+00],
1186            [ 0.0000e+00, -4.1399e-01,  4.7263e-02],
1187            [ 7.8107e-02,  1.5172e+00,  8.4346e-01]]),
        'PassLogTorchIntermediateTempVar1_PassFlattenCallTempVar1': tensor([[ 1.1649e-15,
            7.8107e-02,  1.5172e+00],
```

```
          [ 0.0000e+00, −4.1399e−01,  4.7263e−02],
          [ 7.8107e−02,  1.5172e+00,  8.4346e−01]])}
GPU:
{    'PassFlattenCallTempVar1':  tensor([[6.7582e−15, 6.7582e−15, 1.8113e+00],
          [4.5636e−41, 4.5636e−41, 7.9030e−01],
          [6.7582e−15, 1.8113e+00, 1.0308e−01]], device='cuda:0'),
      'PassLogTorchIntermediateTempVar1_PassFlattenCallTempVar1': tensor([[6.7582e−15,
          6.7582e−15, 1.8113e+00],
          [4.5636e−41, 4.5636e−41, 7.9030e−01],
          [6.7582e−15, 1.8113e+00, 1.0308e−01]], device='cuda:0')}
"""
import torch
torch.manual_seed(0)
torch.cuda.manual_seed(0)
torch.empty_strided((3,3), (1,2))

import torch
torch.manual_seed(0)
torch.cuda.manual_seed(0)
torch.empty_strided((3,3), (1,2)).cuda()
```

Listing 18: New Detected Bug 8

```
"""
API: torch.linalg.cond
Exception Type: VarInconsistentCatch (420)
Error Message:
cond_num on CPU: tensor(1.6336e+08)
cond_num on GPU: tensor(2.9279e+08, device='cuda:0')
"""
import torch
# Create a square matrix
a = torch.tensor([[1., 2., 3.], [4., 5., 6.], [7., 8., 9.]])
# Calculate the condition number
cond_num = torch.linalg.cond(a)
print("Condition number (ord=2):", cond_num)
cond_num_inf = torch.linalg.cond(a, p=float('inf'))
print("Condition number (ord=inf):", cond_num_inf)
```

Listing 19: New Detected Bug 9

```
"""
API: torch.linalg.eigh
Exception Type: VarInconsistentCatch (420)
Error Message: Detail is too long
v on CPU: tensor([[−0.8944,   0.4472],
        [ 0.4472,   0.8944]], dtype=torch.float64)
v on GPU: tensor([[ 0.8944,   0.4472],
        [−0.4472,   0.8944]], device='cuda:0', dtype=torch.float64)
"""
import torch
A = torch.tensor([[1., 2.], [3., 4.]], dtype=torch.float64)
w, v = torch.linalg.eigh(A)
print(w.is_contiguous()) # True
A = torch.tensor([[1., 2.], [3., 4.]], dtype=torch.float64).T
w, v = torch.linalg.eigh(A)
print(w.is_contiguous()) # False
```

Listing 20: New Detected Bug 10

```
"""
API: torch.scatter
Exception Type: VarInconsistentCatch (420)
Error Message:
Result on CPU: tensor([[3, 1, 2],
        [5, 6, 2]])
Result on GPU: tensor([[1, 1, 2],
        [4, 6, 2]], device='cuda:0')
"""
import torch
x = torch.LongTensor([[0, 1, 2], [0, 1, 2]])
src = torch.LongTensor([[1, 2, 3], [4, 5, 6]])
torch.scatter(x, 1, torch.LongTensor([[0, 0, 0], [0, 0, 1]]), src)
```

