# OpenReview forum: "Searching Strengthens Large Language Models in Finding Bugs of Deep Learning Libraries"
_ICLR.cc/2025/Conference — ICLR 2025 Conference Withdrawn Submission_

### Official Review · Reviewer_cHvs · 2024-11-01

**Soundness:** 1
**Presentation:** 2
**Contribution:** 1
**Rating:** 3
**Confidence:** 5

**Summary:**

Fuzzing generates random programs to find software bugs, and rely on three key criteria: rarity, validity, and variety. This paper proposes EvAFuzz, a search algorithm for fuzzing Deep Learning Libraries with LLMs, which focuses on producing rare programs iteratively to detect bugs. EvAFuzz also incorporate specific design components to improve the program validity and variety. To improve validity, it prompts an LLM to correct invalid programs based on execution error messages. To improve variety, it selects the parent programs with a diversity-centric probability distribution. Experimental results show that EvAFuzz can find 9 crash bugs in PyTorch (compared to SOTA’s 7), and has a much higher valid rate and API coverage than prior SOTA.

**Strengths:**

- The paper addresses an important problem. The paper highlights the three key dimensions of fuzzing (rarity, validity, and variety), and attempts to balance the validity-variety tradeoff.
- The approach appears to be broadly applicable, with potential use beyond DL libraries (as shown in the appendix).
- The technique is easy to understand.

**Weaknesses:**

- Lack of novelty.
  - It seems that the approach mainly combines FuzzGPT and TitanFuzz, using a historically bug-triggering program dataset (introduced in FuzzGPT) as the seed pool for an evolutionary algorithm very similar to TitanFuzz. The primary novelties are (1) modifying the scheduling algorithm to enrich diversity and (2) adding a self-repair step for each generated program. These changes appear incremental, as both techniques are well-studied in the fuzzing and program repair literature. Moreover, the benefit of these additional designs is not adequately justified in the experiments (discussed later).
  - Line 291-293: “this approach is in contrast to prior LLM-based fuzzers, which limit seed program selection to those that have the same API as the current selected API”. The statement is misleading. Even FuzzGPT already uses the programs of different APIs to generate programs for a selected API.

- Design flaws in the approach
  - According to section 4.5 (Feedback Scheme), both valid and invalid programs can be selected as few-shot examples (Please correct me if I misunderstood). Is there a specific reason for this choice? Typically, evolutionary algorithms for fuzzing only keep valid programs for further evolution/mutation e.g. [a,b].
  - In Section 3.1, “rarity” is measured by search depth, which seems problematic. Deeper search depth does not necessarily guarantee higher rarity. This is a hypothesis and should be validated by an appropriate rarity metric independent of the search depth.
  - In Section 3.2 and Section 4.4, “rarity” is measured by TitanFuzz’s fitness function. This metric seems to reflect program complexity/diversity rather than the true “rarity” or “edge-case”. This is because it measures the number of unique API calls (which reflects diversity) and the depth of dependency graph (which measures the complexity of API interactions) [b]. A long program with diverse, standard API usage will score high without being rare/edge-case. On the other hand, a short program with a rare edge-code could score low. A better metric might be the code coverage of all generated programs, as API coverage alone is very limited.

- Main argument is not supported by evaluation results.
  - “Rarity” can be measured by code coverage, as many library code paths/branches require rare/edge-case input programs and thus are hard to hit by fuzzers. EvAFuzz has lower code coverage than prior SOTA.


- Insufficient evaluation.
  - Missing efficiency metrics: Efficiency is crucial in fuzzing. It would be beneficial to add a time cost comparison with prior work, especially as self-repair in EvAFuzz could double the time cost per generated program. Prior work [c] suggests that self-repair does not always bring improvement when considering the additional inference cost. If it’s difficult to compare with prior work due to different hardware setup, an ablation study comparing EvAFuzz with or without self-repair under the same time budget would be good.
  - Missing total bug metric: Only crash bugs are reported. The total number of detected bugs - a critical metric for fuzzers - should be reported.
  - Also, the authors should distinguish between false positives and real bugs. The inconsistency shown in figure 4-2 is not a real bug: according to the API documentation [d], when y is negative, the result is implementation-defined. From the appendix, it seems that the authors just listed the raw inconsistency logs, without further inspecting the root cause and verifying if it is a real bug or just false positives.
  - Unfair comparison with baselines. EvAFuzz uses CodeQWen1.5-7B-Chat, a SOTA code generation LLMs, while baselines use older models (InCoder, Codex, and CodeGen) It is unclear how much of EvAFuzz’s improvement comes from the stronger model. It would be more convincing if the authors can run both EvAFuzz and baselines on the same LLMs (either InCoder/CodeGen or CodeQWen1.5-7B-Chat) and compare the results.

- Other minor comments:
  - The citation style is inconsistent. In the first paragraph, the space before the parenthesis is missing.
  - Line 453 `NumGenerated` should be `TargetNum`.



[a] IFuzzer: An Evolutionary Interpreter Fuzzer using Genetic Programming

[b] (TitanFuzz) Large Language Models Are Zero-Shot Fuzzers: Fuzzing Deep-Learning Libraries via Large Language Models

[c] Is Self-Repair a Silver Bullet for Code Generation? ICLR’24

[d] https://www.tensorflow.org/api_docs/python/tf/bitwise/left_shift

**Questions:**

1. Could you clarify why the invalid programs are also used as parent programs (as detailed in the weaknesses)?
2. In section 4.5, it appears the feedback scheme significantly improves all metrics (e.g., valid rate increases from 17.02% to 62.66%, API from 312 to 902). I was wondering if the major improvement over prior work is primarily due to this feedback scheme. It would be interesting to test this hypothesis by running the ablation study in the default setting, i.e., run the full EvAFuzz without feedback while keeping the `TargetNum` in Algorithm 1 unchanged. What would the valid rate and time cost comparison look like?

---

### Official Review · Reviewer_wJk8 · 2024-11-02

**Soundness:** 2
**Presentation:** 2
**Contribution:** 2
**Rating:** 3
**Confidence:** 5

**Summary:**

This paper introduces EvAFuzz, an approach that leverages the power of evolutionary algorithms and large language models (LLMs) to enhance the effectiveness of fuzz testing for deep learning (DL) libraries. By addressing the challenge of balancing test case rarity and validity, EvAFuzz iteratively refines program generation through LLM-guided evolutionary search. The framework incorporates a feedback loop that utilizes execution results to correct invalid programs, further improving the validity of generated tests. Experimental evaluation demonstrates the superiority of EvAFuzz over baselines, as it identifies unique crashes, achieves higher valid test case rates, and expands API coverage in both PyTorch and TensorFlow. EvAFuzz uncovered about 10 bugs within the tested DL libraries.

**Strengths:**

+ Addresses a Significant Problem: The paper tackles an important problem in deep learning library testing: generating rare and valid test cases to uncover bugs.
+ Innovative Approach: EvAFuzz introduces a feedback-driven approach that effectively balances test case rarity and validity.
+ Strong Empirical Results: The proposed method outperforms baseline tools in terms of API coverage and validity rate.
+ Real-World Impact: Contributes to real-world bug detection for the widely-used DL libraries.

**Weaknesses:**

- Limited Novelty: The core idea of using LLMs to generate test cases shares similarities with existing approaches like FuzzGPT. While the feedback loop is a valuable contribution, it may not be sufficiently distinct to warrant high novelty.

- Unclear Experimental Setup: The paper lacks clarity regarding the specific fuzzing budget allocated to EvAFuzz during evaluation. This omission hinders a comprehensive understanding of the performance gains.

- Inconsistent Coverage Results: Despite the claim of improved coverage, the reported line coverage numbers for EvAFuzz are lower than those of the baseline tools in both PyTorch and TensorFlow. This inconsistency raises questions about the effectiveness of the approach in exploring rare code paths.

- Limited Bug Discovery: While the paper highlights the detection of over 10 bugs, further details about the severity, novelty, and impact of these bugs are missing. Plus the number of detected new bugs is limited compared to baselines. Additionally, information on the number of confirmed previously unknown bugs and fixed issues would provide a clearer picture of the practical impact.

**Questions:**

1. Why does EvAFuzz achieve lower line coverage compared to baseline methods?

2. What types of bugs is EvAFuzz specifically designed to detect? Is it aimed at valid programs that might trigger mis-compilations, or at rare invalid programs that cause system crashes or unexpected passes?

3. What are the statistics of the newly detected bugs with respect to categories like new, confirmed, and fixed bugs? Additionally, why does EvAFuzz contribute a relatively limited number of new bugs compared to other methods?

---

### Official Review · Reviewer_EcAK · 2024-11-03

**Soundness:** 2
**Presentation:** 2
**Contribution:** 2
**Rating:** 3
**Confidence:** 3

**Summary:**

This paper present EvAFuzz -- a technique for fuzzing Deep Learning libraries. EvAFuzz attempts to generate effective fuzzing inputs by producing more valid and rare programs. EvAFuzz works through a search algorithm to select more rare programs as few-shot examples inputs as well as feedback mechanism to correct any invalid programs to increase validity. The results show that EvAFuzz can achieve higher variety in terms of API coverage as well as increasing the validity rate compared with prior approaches.

**Strengths:**

- The work tackles an important area of bug finding
- The approach and technique proposed by the authors is simple and intuitive to understand
- The additional experiments on different libraries (i.e., NumPy) and using other models is comprehensive

**Weaknesses:**

- **Lack of evidence for rarity**:
	- The authors define a rare program as "program that covers a specific edge case", however I do not see any evidence in the paper that seems to evaluate that comprehensively
		- While the authors mentioned coverage as an evidence, it is only line coverage from my understanding. Would edge coverage instead not be a more accurate estimation of rarity to see if the newer programs generated can cover new edges (i.e., execution paths)?
	- Figure 2: rarity using the fitness function score is just not an accurate measure
		- Since higher fitness function score is used to for selection of the examples in the few-shot prompting setup of course choosing higher fitness function scored programs will lead to more higher scored programs (by copying previous examples). The fitness function as explained in Equation 1 seems to be a very crude way of measuring rariety as it only focuses on program rariety and not rariety with respect to the pathes and executions in the underlying library being tested
		- Also, in Section 4.7, the authors refer to the fitness function as "our" fitness function, but it seems it was directly reused from (Deng 2023)? Did the authors make additional changes to the computation of the fitness score?
	- Overall, the definitely of rarity does not match the evidence demonstrated in the paper
- **Lack of bug finding**:
	- The authors did not provide updates to the exact number of bugs found in both PyTorch, TensorFlow, and later on other data science libraries
	- Crashes by itself does not indicate bug finding prowess as many of these crashes could be labeled as unimportant by the developers or can already be found by prior techniques. I would suggest the authors to actually report the bugs to the developers and observe their responses.
	- Furthermore, how many of these crashes are actually important bugs are hard to tell
- **Questionable feedback scheme**:
	- The scheme proposed by the authors aims to get more valid programs as seeds for few-shot examples (in Section 4.5: "However, without the feedback scheme, invalid programs dominate as seed programs, increasing the likelihood of generating more invalid programs.") however, can we not just first filter out any invalid programs and only use valid programs as seeds?
		- in fact, from a quick glance at the prior work (Deng 2023) which the authors got the fitness function from, it seems they only consider the valid programs when doing selection for next round of program generation.
		- then the question is why not follow the original valid selection mechanism?
	- The feedback scheme proposed also seems to incur huge amount of time, since the approach requires an additional update to each of the invalid programs
		- this time cost is not measured, an alterative approach and a more fair comparison (instead of the w/o feedback baseline in 4.5) would be to keep the time consistent and generate more programs without using feedback
- **Unclear parameters and statistics**:
	- In Section 4.2 the authors only briefly highlighted what the default parameters are for the technique
	- However, I have no idea what the other important parameters are (e.g., number of target APIs, number of targeted program generates
	- Furthermore, there are also no statistics for the time cost of fuzzing which is extremely important for efficiency which the authors do not seem to consider




**Minor**:
- In text citations seems to be missing spaces before bracket in certain cases (e.g., the citations in paragraph 1 in Introduction, some citations in Section 2.2)
- Line 353: Firstly, We -> we
- Line 368: Secondly, We -> we

**Questions:**

- Curious why the authors chose to use temperature 0.8, intuitively 1 would make more sense to generate more diverse programs? (i.e., are there any additional experiments done to pick the best temperature for fuzzing and generating rare programs)
- Can the authors expend on the feedback loop used in the technique? For example why not select only the valid programs for the next round?
- What is the run time of the hyperparameters chosen by the authors? it would be unfair compare these fuzzing strategies under widely different time frames.
- Can the authors comment on the bug finding process for this paper, did the authors submit any issues for the developers?
- If possible please also address the weakness highlighted in the weakness section

---

### Official Review · Reviewer_FfHD · 2024-11-04

**Soundness:** 2
**Presentation:** 3
**Contribution:** 2
**Rating:** 5
**Confidence:** 5

**Summary:**

The paper introduces **EvAFuzz**, a novel framework combining Evolutionary Algorithms (EA) and Large Language Models (LLMs) to enhance fuzz testing for deep learning libraries like PyTorch and TensorFlow. EvAFuzz focuses on improving three critical dimensions of fuzzing: rarity, validity, and variety of generated programs. Through the combination of EA and LLM, EvAFuzz generates rare programs that trigger bugs in deep learning libraries more effectively, corrects invalid programs via a feedback mechanism and increases program diversity. The experimental results compare the performance of EvAFuzz with the previous state-of-the-art fuzzing approaches in terms of bug detection, API coverage, and program validity rates.

**Strengths:**

The paper explores an interesting direction, following the intersection of LLM capabilities with evolutionary computation, which is promising due to the ability of the former to perform generation of high variety (under the right conditions) drawing synergy from the latter's optimisation capabilities. Specifically, search is an understudied EA subdomain, which further increases the attractiveness of the proposed research and its potential to improve the debugging processes for highly complex software.

**Weaknesses:**

## Link to past works
Previous works, which are also cited by the authors, have explored integrating LLMs with EA for fuzzing. While the authors do not explicitly claim novelty in this sense, the choice of name for the proposed method implicitly has the reader assume so. Therefore, the paper does not adequately position itself within the existing body of research that combines LLMs and EAs for similar purposes, due to the lack of direct comparison of the proposed methodological developments to the prior works.

## Rarity
The concept of rarity is central to the paper's methodology, yet its definition in Section 3.1 lacks clarity. Equation (1), which is intended to represent rarity, appears to deviate from the initial description and seems more aligned with measuring program complexity rather than rarity, based on its description provided by the cited authors that proposed this equation. The paper does not provide evidence or justification that a more complex program is inherently rarer. This ambiguity weakens the validity of the experimental results and the conclusions drawn from them.

## Methodology description
Significant parts of the methodology are not described in sufficient detail, which hinders reproducibility. Key components, such as the implementation details of the evolutionary algorithm (particularly the `Update()` function) and the feedback mechanism, are inadequately explained and do not provide the expected transparency for a publication of this level.

## Evaluation
The paper does not address how the proposed method performs with weaker LLMs or how the required computational resources differ from the benchmark model. Since the success of the approach may rely heavily on the power of the underlying LLM and the allocated compute budget, the absence of this analysis leaves questions about the source of the reported fuzzing results improvement unanswered.

The authors mentioned that the default value for **MaxRetry** is 1 which means that the LLM has only 1 shot to correct the generated program if it fails to execute, and they did not conduct experiments with different values of this parameter which could highly influence some metrics like validity. Setting the parameter **NumPrograms** to 2 => prompting the LLM with only 2 seed programs would probably have a negative impact on the variety metric, and no other values were tested.

## Other comments
Figure 1 does not show the two oracles used, but only one. The text then states that the authors employ two types of oracles, but then they exclude one of the oracles from comparison.

Section 4.1 Metric is poorly written, providing insufficient details about the purpose, justification and implementation of the proposed experimental metrics.

Section 4.4 seems redundant, only following a relatively obvious argument about the validity-rarity trade-off, while ignoring the undescribed peculiarity about the Score dynamics, which sharply drops below the initial level following the first rise. Moreover, the algorithm seems to tend to 0% validity rates towards the end of the evolution, which is likewise not discussed in the paper.

**Questions:**

### Rarity
1. Where is the definition of rarity in Section 3.1 produced from?

2. Can you prove that a complex program is inherently rare?

3. How is the evolutionary algorithm implemented? I.e., what is the Update() function? If it is based on one of the cited EA fuzzing approaches, these have to be stated with details on the reused and/or adapted part of the algorithms.
### Validity
4. How does the validity-enhancing feedback algorithm perform in terms of computational costs? I.e., can you prove that for the purpose of a the creation of a **higher total absolute number of valid programs** the additional compute costs due to A. the inclusion of this feedback mechanism is more efficient than a similar compute cost increase spent on B. a naive generation of a higher number of programs?

5. Do you observe that the feedback mechanism adapts programs to different APIs due to how API and Program selection are independent? Does this draw inspiration from previous works on applications of transfer learning to fuzzing?
### Results comparison against baseline
6. What is the issue with the quantification of the inconsistency bugs?
7. How would different combinations of the parameters **MaxRetry** and **NumPrograms** effect the performance of the algorithm over the 3 different metrics (especially variety and validity)
#### LLM strength
8. How does EvAFuzz perform when used with a weaker LLM, comparable to the InCoder 1.3B used by authors of TitanFuzz?

9. How does EvAFuzz perform under fixed computational resources as compared against alternatives?

---

### Note · Authors · 2024-11-25

**Comment:**

Dear Reviewers,

We are authors of the paper "Searching Strengthens Large Language Models in Finding Bugs of Deep Learning Libraries" (ID: [9407]). We sincerely appreciate your valuable feedback and insights. After careful consideration, we have realized the shortcomings in our manuscript and have decided to withdraw it for further refinement. We are committed to addressing all your comments and improving the paper to meet the high standards. Thank you once again for your guidance and support.

**Withdrawal Confirmation:**

I have read and agree with the venue's withdrawal policy on behalf of myself and my co-authors.